# Effect of Ce Addition and Heat Treatment on Microstructure Evolution and Tensile Properties of Industrial A357 Cast Alloy

**Yanfeng Wang [1], Qian Liu [1], Zheng Yang [1,2,*], Changming Qiu [1] and Kuan Tan [1]**

1   College of Mechanical Engineering, North China University of Science and Technology,
    Tangshan 063210, China; wyf@ncst.edu.cn (Y.W.); ncst_qian@163.com (Q.L.); qcm@ncst.edu.cn (C.Q.);
    xrcjh9rqmcx@163.com (K.T.)
2   Technology center, Delong of Tangshan Iron & Steel co., LTD, Tangshan 063210, China
*   Correspondence: ncsth9r@163.com; Tel.: +86-159-332-523-85

**Abstract:** The effects of adding different Ce contents (0–0.32 wt.%) on the microstructure, mechanical properties, and fracture morphology of industrial A357 cast alloy in as-cast and T6 heat treatment were studied. The main purpose of this study is to improve the microstructure stability and tensile properties of industrial A357 cast alloy. The microstructural analyses indicate that the addition of Ce causes refinement of the $\alpha$-Al primary phase for the reason that the formation of intermetallic compounds containing (AlSiCeMg) elements enriches the front of the solid–liquid interface, which causes an increase in constitutional undercooling. Simultaneously, the addition of Ce also affected the characteristics of eutectic Si particles, which make its morphology change from acicular structures into fragmented and spheroidized. This is mainly due to the formation of Ce-rich precipitates during solidification, which increase the constitutional undercooling and suppress the nucleation of the eutectic Si particles, resulting in the change of eutectic Si characteristics. Moreover, the needle-like morphology of a Fe-containing intermetallic is transformed into $\alpha$(AlSiFeCe) phase containing rare earth Ce when part of the Ce atoms entered $\beta$(Al5FeSi) phase compounds. The tensile properties of the modified alloys were improved in the as-cast and T6 heat treatment as a consequence of simultaneous refinement of both secondary dendrite arm spacing and grains and the improvement of eutectic Si particles and Fe-containing intermetallic morphology. The fracture surface of the modified alloy has more dimples than the unmodified alloy, which indicates that the main fracture pattern of the modified alloy is dimple fracture caused by the crack of eutectic Si particles. The optimal percentage of Ce in industrial A357 cast alloy was determined to be 0.16 wt.% according to the change of microstructures structure and mechanical properties. These experimental results provide a new basis for adding rare earth Ce to improve the performance of parts in the actual production of industrial A357 cast alloy.

**Keywords:** industrial A357 cast alloy; rare earth Ce; microstructure; tensile properties; intermetallic compound; heat treatment

## 1. Introduction

The industrial A357 cast alloys has been progressively used to critical structural in aerospace and automobile industries for reasons of numerous advantages such as outstanding castability, corrosion resistance, etc. [1,2]. It is a known fact that the tensile properties of the Al-Si cast alloy are mainly influenced by the coarse dendritic structures, eutectic Si morphology, and intermetallic compounds [3,4]. In previous literature, extensive efforts have been dedicated to grain size refinement through chemical modification, such as addition of Al-Ti-B, Al-Ti-C, etc. In addition, the addition of

trace Na proved to be an effective modifier for the transformation of eutectic Si morphology from needle shape to fiber structure [5,6]. However, the addition of chemical elements to refine the α-Al primary phase and eutectic Si modification still several drawbacks. For example, the poisoning effect of chemical elements between B and Sr will reduce the efficiency of grain refinement and eutectic Si modification [7]. The rare earth elements have special activity compared with chemical modifiers [8] and have a good effect on the secondary dendrite arm spacing (SDAS), grain size refinement, eutectic Si modifiers, and hydrogen removal, as a consequence the tensile properties of the alloy are improved. Furthermore, rare earth elements are an excellent modifier for cast aluminum alloys because of their relatively long-lasting properties and remelting stability [9,10].

In the past several years, many investigations have focused on the effects of rare earths such as Eu, Y, Sc, La, Er, and Gd on the grain size, SDAS eutectic Si, and mechanical properties of A356 casting alloys. Generally, different types of rare earth elements have different modification effects. Li et al. [11] have reported that when 0.05 wt.% Eu was added to the Al-5Si alloy, the multiply twinned Si particles were formed in virtue of the precipitate of Eu-rich atoms along the <112> Si growth direction of Si and precipitate at the intersection of two {111} Si twins in eutectic Si. Dong et al. [12] showed that the addition of Sr-Y modifies the eutectic Si morphology of the A356 alloy. They concluded that due to the formation of Al3Y compounds, which increased the α-Al crystal nucleus size, the grain size was decreased. The eutectic Si morphology in the A356 alloy was transformed into granular or flaky after the addition of Sr-Y composite modifier. Most of the eutectic silicon was further converted into smaller particles when subjected to T6 heat treatment. Pandee et al. [13] showed that the α-Al primary phase was refined after the addition of Sc; this is because the precipitate of Al3Sc can be used as heterogeneous nucleation point. Nevertheless, its effect on eutectic Si morphology is much weaker. Shi et al. [14] have reported that the effects of Er on the microstructure and tensile properties of A356 alloy in the as-cast state were comprehensively studied. In their opinion, the addition of 0.3 wt.% Er to A356 alloy has a significant effect on the morphology of the eutectic Si and α-Al primary grain refinement. Song et al. [15] studied the effect of (Pr + Ce) on the microstructure and mechanical properties of a Al-7Si-0.7Mg alloy. The acicular eutectic Si changed into a rose-like shape, with a reduced grain size and high spheroidization. In addition, the constitutional undercooling caused by the addition of (Pr + Ce) effectively refines the grain size and improves the tensile properties of the alloy. Kang et al. [16] found that the separate addition of Mg or Ce in Al-7Si-0.3Mg-0.2Fe alloy can cause grain and eutectic Si refinement, while the simultaneous addition of Mg and Ce has a more obvious grain refinement and excellent eutectic Si modification. They suggest that the proper combination of Mg and Ce has a beneficial effect on grain size and eutectic Si morphology of Al-Si casting alloy production. Nevertheless, the influence of Ce on the α-Al primary, eutectic Si modification, and mechanical properties of A357 casting alloy has not been reported in the literature, and it has higher quality parameters than A356 cast alloy [17].

Industrial A357 aluminum alloy has been widely used in practical production. Therefore, the experimental results of studying the effect of rare earth Ce on the properties of industrial A357 aluminum alloy can provide a more meaningful reference for practical production. In this study, the rare earth Ce with different content was added to the industrial A357 casting alloy. The effect of rare earth Ce on the microstructure and mechanical properties of industrial A357 casting alloy in as-cast and T6 conditions was studied and the optimal content of rare earth Ce was obtained. What is more, the mechanism of rare earth Ce on the morphology of α-Al primary phase size and intermetallic compounds and eutectic Si modification of industrial A357 alloy was explained to clarify the effect of rare earth Ce on the mechanical properties of industrial A357 alloy.

## 2. Experimental Procedure

The chemical composition of the industrial A357 alloy used in this research is presented in Table 1. First, the industrial A357 aluminum ingots were melted in the resistance furnace crucible, when the temperature of the metal solution reached 750 °C. Different amounts of Al-10 wt.% Ce intermediate

alloys were added to the molten melt to form five sets of samples with different Ce contents (0%, 0.08%, 0.16%, 0.24%, and 0.32%), which are named as C0, C1, C2, C3, and C4, respectively. It should be emphasized that modifiers such as Sr or Na were not added to the alloy. During the melting of the added master alloy, the slag was skimmed and the molten metal was refined using argon gas for 10min. Subsequently, when the melting temperature was then decreased to 720 °C, it was poured with different compositions into a metal mold coated with ZnO, and the preheating temperature of the mold was 250 °C. The samples in one group were subjected to solution T6 treatment. The samples were heated to 540 ± 5 °C for 5.5 h in a resistance furnace, and then quenched in water at 80 °C for no less than 5 min. The samples then underwent aged treatment at 165 ± 5 °C for 4 h and were eventually cooled in air.

**Table 1.** Chemical compositions of research alloys (wt.%).

| Element | Si | Fe | Cu | Mn | Mg | Zn | Ti | V | Al |
|---|---|---|---|---|---|---|---|---|---|
| Content (wt.%) | 7.17 | 0.135 | 0.004 | 0.002 | 0.474 | 0.004 | 0.142 | 0.023 | Balance |

The metallographic samples were polished according to the standard metallographic procedure and then etched with 0.5 wt.% hydrofluoric acid reagent for the analysis of the microstructure. The average grain size, SDAS, and the aspect ratio of the Si particles of the specimens were observed using a metallographic microscope, which was evaluated using image analysis software (LAMOS Expert). The average grain size, SDAS, and eutectic Si morphology were used as the indexes of microstructure improvement and eutectic modification, respectively. According to the average length of Si particles as well as the average width of Si particles, the aspect ratio of Si particles was calculated. The equivalent particle size of α-Al phase is defined according to the following equation,

$$D = 2(A/\pi)1/2 \tag{1}$$

where $A$ is the average area of the grain.

Scanning electron microscopy (SEM) (Hitachi, Tokyo, Japan) and energy spectroscopy (EDS) analyzers (Hitachi, Tokyo, Japan) were utilized to obtain the morphology and chemical data of the phase and observe the fracture morphology. The phase composition of the specimens was analyzed using X-ray diffraction (XRD) with a D/MAX2500PC (Rigaku, Tokyo, Japan), whose radiation target was Cu-Ka ($\lambda$ = 1.54056 Å). The XRD spectrum was obtained in the 2θ range from 10 to 90° at a scanning speed of 2°/min and with a step size of 0.02°. The mechanical properties of the samples were tested at room temperature by using the INSTRON universal testing machine (Instron, Boston, MA, USA) with a loading speed of 1.0 mm/min. The tensile sample is processed according to the GB/T1173-2013 standard, and the sample size is shown in Figure 1.

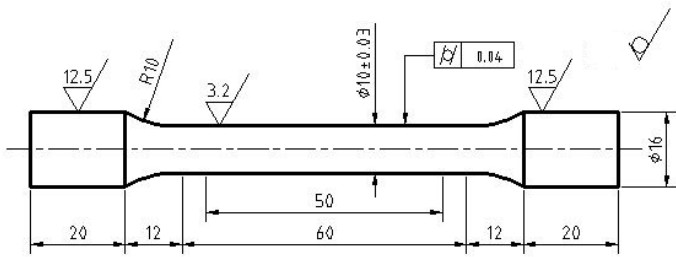

**Figure 1.** Diagram of tensile samples.

## 3. Results and Discussion

### 3.1. Effect of Ce on the Refinement of α-Al Grains

Figures 2 and 3 show the optical microstructure of the industrial A357 alloy with different Ce content in as-cast and T6 heat treatment conditions. The grain size and SDAS measurements, resulting in α-Al primary phase with different Ce addition under the condition of as-cast and T6 heat treatment, are shown in Figure 4. It can be seen from the figure that the addition of Ce effectively reduced the grain size and SDAS value of the α-Al primary phase. As shown in Figures 2a and 3a, coarse dendrites and nonuniform size were observed in the microstructure of the without addition rare earth alloy in as-cast and T6 heat treatment. In contrast, the coarse α-Al primary dendrites were refined by adding rare earth Ce to industrial A357 alloy. In the as-cast and T6 samples with the addition of 0.08 wt.% Ce (Figures 2b and 3b), the grain size and SDAS value have significantly changes, and were decreased to 45.07 μm, 24.67 μm, 39.36 μm, and 22.46 μm, respectively, compared with the samples without addition of rare earth alloy. With the increase of Ce content to 0.16 wt.% (Figures 2c and 3c), the proportion of coarse dendrites decreased significantly, and fine equiaxed grains dominated, with only individual elongated grains. Most of the grains are thin and round, and the distribution is relatively uniform. At this time, the grain size is 39.54 μm and 21.16 μm and the SDAS value is 28.29 μm and 17.32 μm, under the conditions of as-cast and T6 heat treatment, respectively. The reductions in the size of grain are 50.65% and 42.41% and the SDAS value decrease 54.93% and 45.99%, respectively, compared with the unmodified alloy. which obviously achieves grain refinement. However, when Ce content was further increased to 0.32 wt.% (Figures 2e and 3e), the morphology and size of the α-Al primary phase began to deteriorate. It was found that the coarse dendrite structure and its shape changed irregularly. The grain sizes are 63.8 μm and 49.21 μm and the SDAS values are 39.37 μm and 26.17 μm, respectively, under the condition of as-cast and T6 heat treatment (Figure 4).

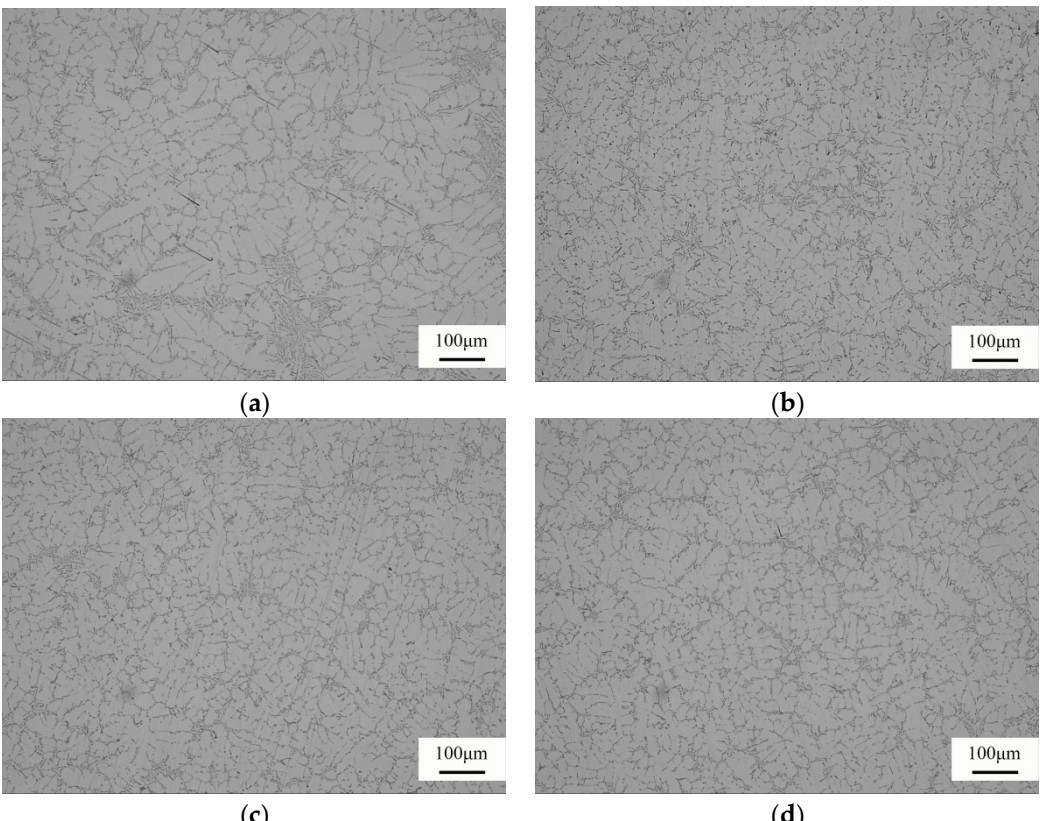

(a)  (b)

(c)  (d)

**Figure 2.** *Cont.*

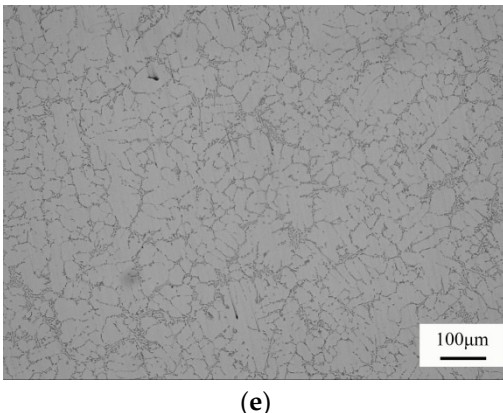

(**e**)

**Figure 2.** As-cast metallographic diagram of A357 alloy with different Ce content. (**a**) 0%, (**b**) 0.08%, (**c**) 0.16%, (**d**) 0.24%, and (**e**) 0.32%.

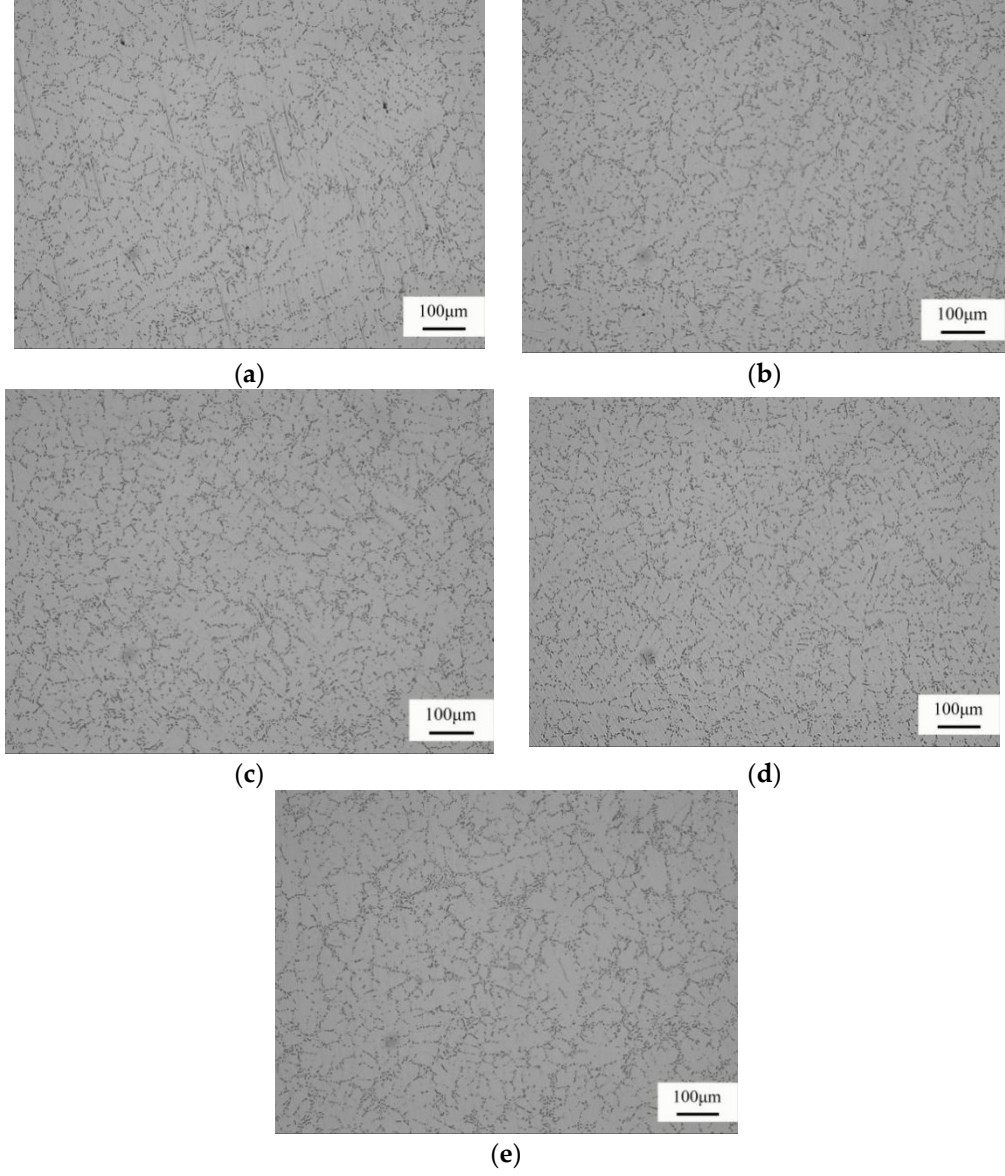

**Figure 3.** T6 metallographic diagram of A357 alloy with different Ce content. (**a**) 0%, (**b**) 0.08%, (**c**) 0.16%, (**d**) 0.24%, and (**e**) 0.32%.

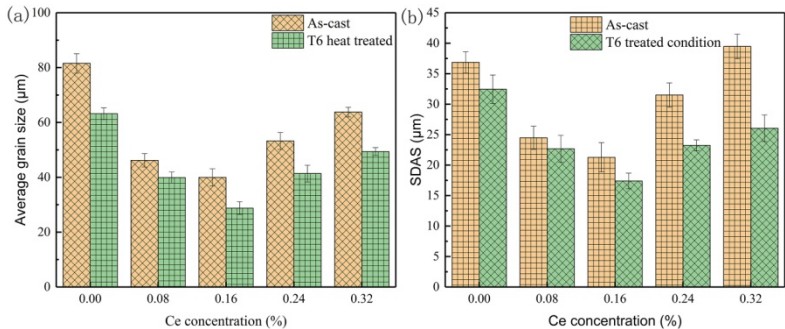

**Figure 4.** Statistical results of average grain size and secondary dendrite arm spacing (SDAS). (**a**) Grain size, (**b**) SDAS.

In the process of heat treatment, the aging treatment mainly affects the precipitation of the second phase due to the lower temperature of the aging treatment. However, the solution treatment may also affect the α(Al) grain of the excess phase when it is fully dissolved into the solution. Shi et al. [18] studied the effect of solution treatment on the microstructure and mechanical properties quasi eutectic Al-Si alloy. They believe that as the solution temperature rises during the heat treatment process, the growth rate of α(Al) grains of the ZAlSi12CuMgNi aluminum alloy will also increase. Zeng et al. [19] found that the grains of 7055 aluminum alloy are uniform and fine through metallographic observation after solid solution treatment. Wei et al. [20] found that the solid solution temperature plays a decisive role in the grain size of the as-extruded WE43 magnesium alloy. E. Fan et al. [21] believe that T6 solution aging can increase the structure density of AC4B alloy castings and reduce the DAS of (Al) grains. He et al. [22] found through experiments that after heat treatment, the coarse α-Al primary dendrites in the A356 aluminum alloy structure can be transformed into small cell crystals and spherical crystals. Although the solution treatment will not have a significant effect on the α(Al) grains, it may have a certain effect on its morphology under certain conditions.

To a large extent, the grain size depends on the effective nucleation position and the degree of under cooling during the solidification of metal solutions. Based on the ternary phase diagram of Al-Si-Ce, the eutectic reaction of Ce takes place at 621 °C [23], and the reaction equation is L→ α-Al + $Al_{11}Ce_3$ [24]. Moreover, due to the large electronegativity difference between Ce and Al, $Al_{11}Ce_3$ compounds are easily formed [25]. According to the Bramfitt mismatch theory, the lattice mismatch degree of $Al_{11}Ce_3$ and α-Al is 7.19%, which is between 6% and 12% [26,27]. Therefore, it is considered to be an effective heterogeneous nucleation site. In addition, $Al_{11}Ce_3$ (a = 0.4395 nm) is very to the close to the lattice constant with α-Al (a = 0.4049 nm) [28]; therefore, $Al_{11}Ce_3$ can be used as a heterogeneous nucleation point when α-Al solidifies, resulting in a large number of crystalline cores were formed in the melt. This resulted in the decrease of the size of SDAS and grain of the α-Al primary phase. The XRD pattern of A357 modified alloy also confirmed the existence of the above eutectic reaction products, as shown in Figure 5; although Ti is present in the A357 alloy. However, the effect of Ti on the experimental results is not obvious, as the Ti content of the alloy used in this article is 0.142%. The literature shows that the effect of TiAl3 on the heterogeneous nucleation of (Al) grains and the inhibition of Ti on grain growth were observed with the addition of Ti. However, when adding 0.2% of Ti, the grain refinement of aluminum alloy is not obvious [29].

However, when Ce was added in excess, the grain size instead becomes larger. The phenomena of precipitation and segregation of intermetallic compounds occur when the composition of Ce is high, in that the density of the compound $Al_{11}Ce_3$ (4.123 g/cm$^3$) is much higher than that of pure aluminium [26], as a result of the number of nuclei is reduced and the effect of grain refinement is not satisfactory. On the other hand, the concentration of solutes at the solid–liquid interface is relatively higher due to the fact that the maximum solid solubility of Ce in α-Al is 0.05%, the distribution coefficient Ko < 1 [30], and these solute atom concentrations are much higher than the overall average concentration. Therefore, during the process of alloy solidification, with the formation of α-Al, the Ce

atoms will continuously diffuse to the solidification front, and there will be a transition region of the solute concentration gradient near the solid–liquid interface [31,32]. The concentration of solute atoms is highest at the interface near the α-Al, which is caused by the enrichment of a large proportion of Ce atoms at the interface of α-Al and liquid. This provides favorable conditions for the formation of intermetallic compounds containing one or more elements such as Al, Si, Ce, Mg, etc. at the solid–liquid interface. However, the melting points of these intermetallic compounds are higher than the temperature of the eutectic reaction. These intermetallic compounds do not melt at the solid–liquid interface, which effectively inhibits the continued growth of the α-Al primary phase, as shown in Figure 6a. As shown in Figure 7, many intermetallic compounds containing Al, Si, Ce, and Mg elements were found in sample C2. Moreover, the enrichment of Ce at the front of the solid–liquid interface causes undercooling of the structure [16], which can further refine the grain size to a certain extent.

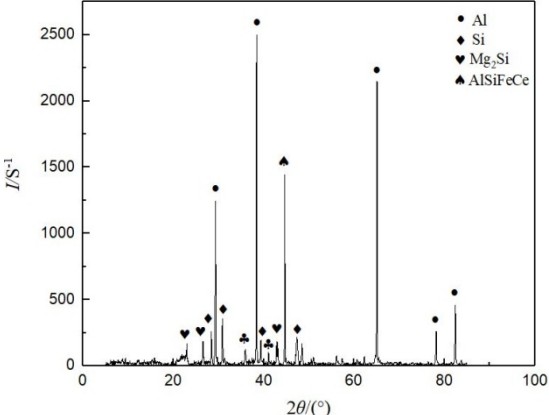

**Figure 5.** XRD diagram of alloy sample.

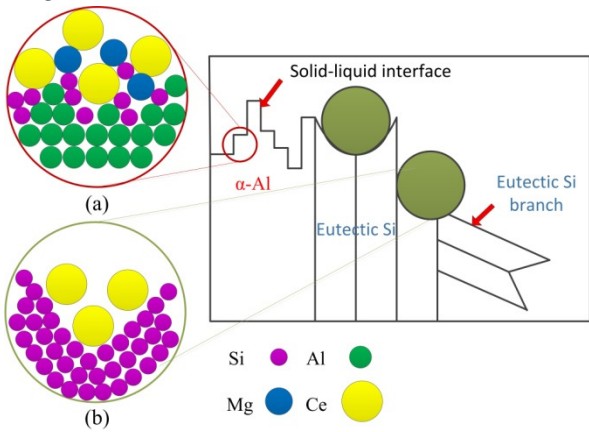

**Figure 6.** Schematic of grain refinement and modification process. (**a**) (Al, Si, Ce, Mg) intermetallic compounds, (**b**) Enrichment of Ce atoms.

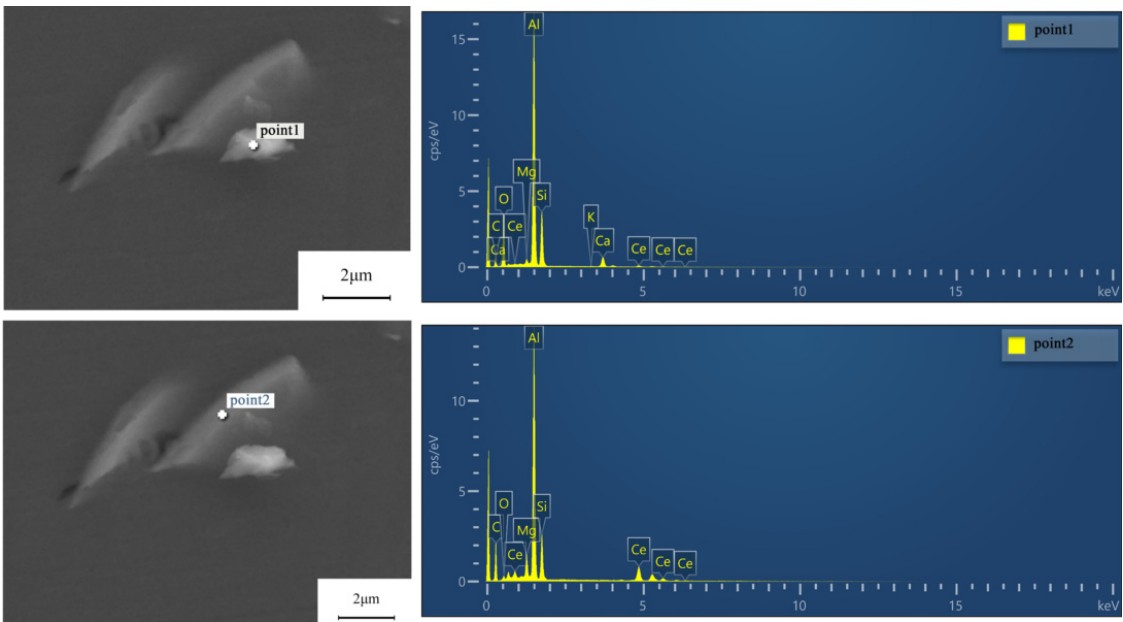

**Figure 7.** Ce-rich intermetallic compounds (Ce-IMCs). (**a**) Point 1 and EDS analysis, (**b**) Point 2 and EDS analysis.

### 3.2. Effect of Ce on the Eutectic Si Morphology

Figures 8 and 9 show the eutectic structure of Si under an optical microscope with different Ce additions in the as-cast and T6 heat treatment. The eutectic Si in A357 Alloy without Ce added under the as-cast condition presents a coarse and plate-like morphology (Figure 8a). Through T6 heat treatment, the plate-like eutectic Si phase was destroyed and some short rods were displayed in the rare earth alloy without addition (Figure 9a). However, this is not sufficient for improving the morphology of eutectic Si. In contrast, when 0.08 wt.% Ce was added (Figure 8b), it can be noticed that the eutectic Si size and the interlamellar spacing were decreased; however, the eutectic Si remained plate-like. Therefore, the eutectic Si particles in the modified alloy with rare earth Ce were obviously spheroidized and evenly distributed under the condition of T6, as the spheroidizing efficiency of eutectic Si particles is mainly determined by its initial size, as shown in Figure 9b-e. In the sample with 0.16 wt.% Ce, the morphology of eutectic Si was further improved, showing a granular structure (Figure 8c). With the further increase of the rare earth content, the eutectic Si particles began to deteriorate, and plate-like morphology was observed.

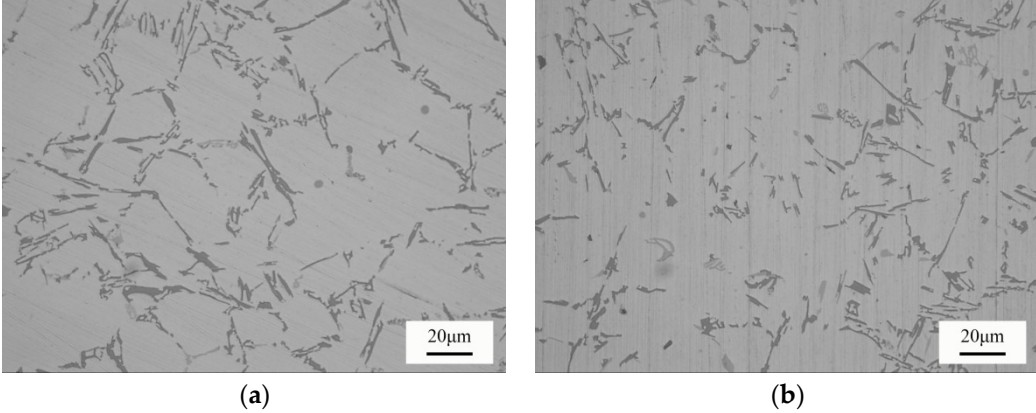

(**a**)　　　　　　　　　　　　　　　　　　(**b**)

**Figure 8.** *Cont.*

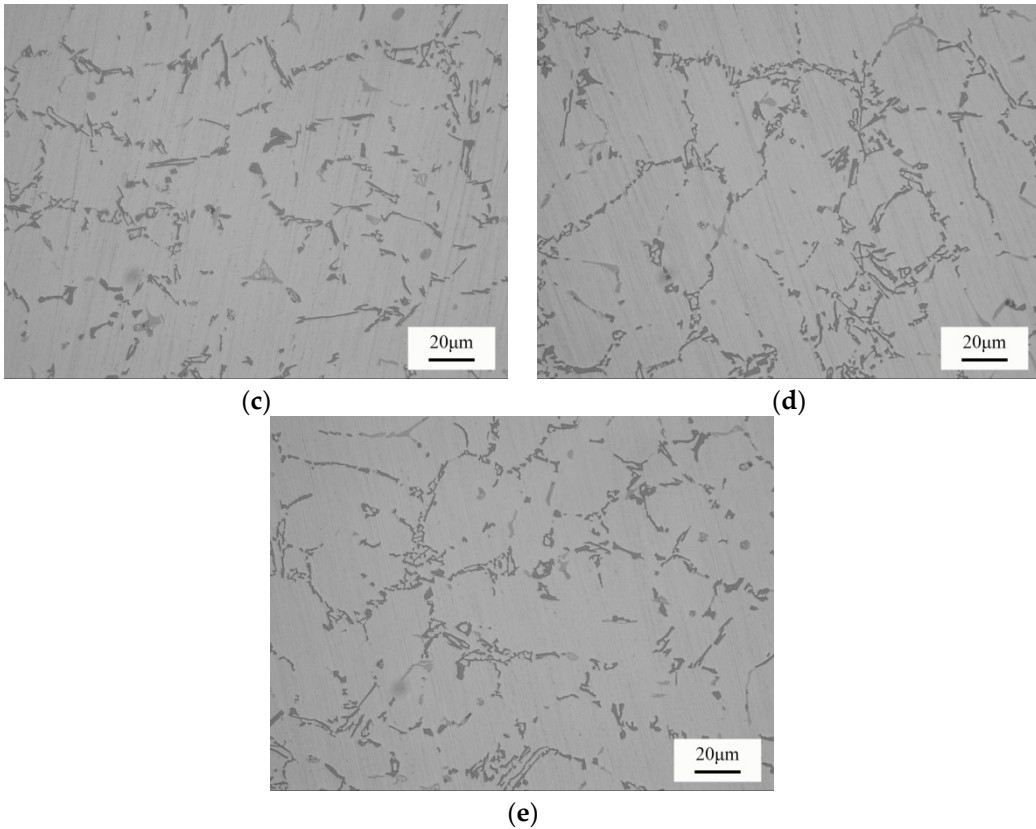

**Figure 8.** As-cast high magnification light optical micrographs of A357 alloy with different Ce content. (**a**) 0%, (**b**) 0.08%, (**c**) 0.16%, (**d**) 0.24%, and (**e**) 0.32%.

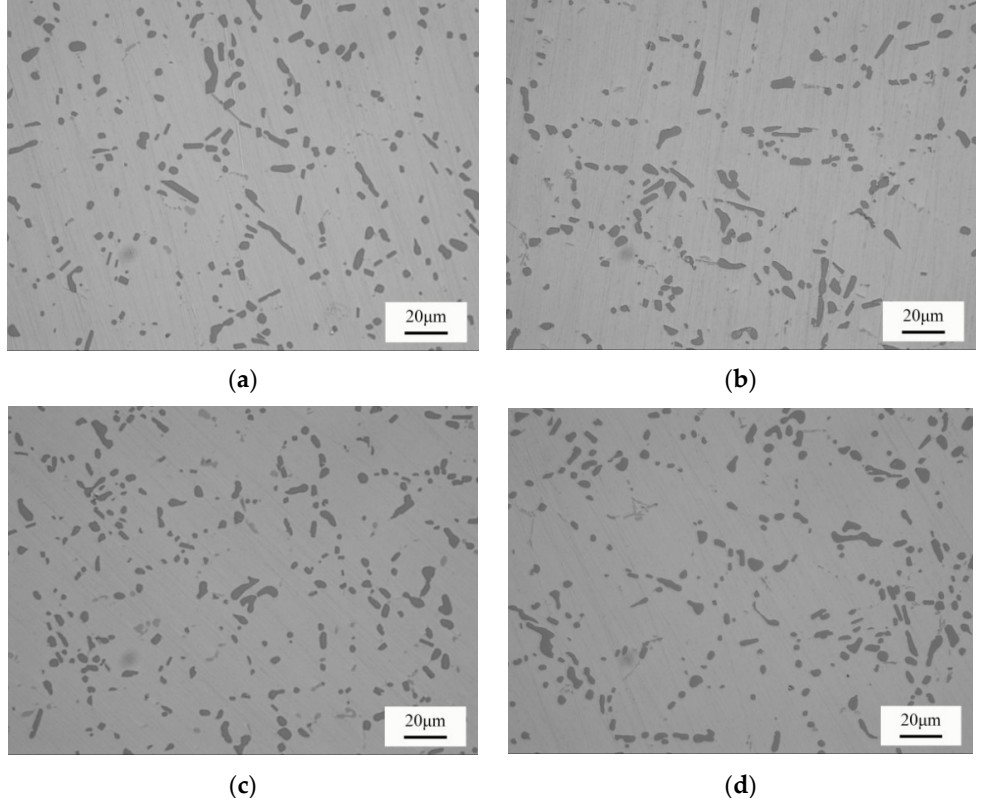

**Figure 9.** *Cont.*

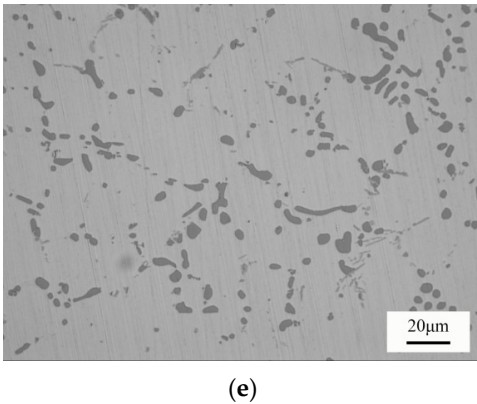

(e)

**Figure 9.** T6 magnification light optical micrographs of A357 alloy with different Ce content. (**a**) 0%, (**b**) 0.08%, (**c**) 0.16%, (**d**) 0.24%, and (**e**) 0.32%.

Table 2 presents the measurement results of eutectic Si particles characteristics including average length, average width, and aspect ratio of A357 alloy with different Ce additions. It can be found that the eutectic Si in the C0 alloy has a relatively high average length and aspect ratio. These morphological indices of eutectic Si particles in the modified alloy are decreased with the addition of Ce. Under the conditions of as-cast and T6, the average length of eutectic Si of the C2 modified alloy decreased by 52.49% and 43.12% and aspect ratio decreased by 72.06% and 76.35%, compared with that of C0 alloy. However, further addition of Ce does not facilitate any further spheroidization.

**Table 2.** Eutectic silicon parameters of different Ce content A357 alloy.

| Condition | Alloy | Si Average Length (μm) | Si Average Width (μm) | Si Aspect Ratio |
|---|---|---|---|---|
| As-cast | C0 | 45.57 | 3.06 | 14.89 |
| | C1 | 24.21 | 4.38 | 5.53 |
| | C2 | 21.65 | 5.21 | 4.16 |
| | C3 | 30.49 | 4.69 | 6.50 |
| | C4 | 34.02 | 4.13 | 8.24 |
| T6 | C0 | 24.07 | 2.82 | 8.54 |
| | C1 | 13.69 | 5.32 | 2.57 |
| | C2 | 10.98 | 5.43 | 2.02 |
| | C3 | 16.59 | 4.93 | 3.37 |
| | C4 | 19.46 | 4.29 | 4.54 |

Regarding the modification of eutectic Si, the addition of Ce can significantly reduce the aspect ratio and length of eutectic Si in as-cast conditions. In current theories of eutectic Si growth models, Impurity Induced Twinning (IIT) and Twin-Plane Reentrant Edge (TPRE) are generally widely accepted [33,34]. In the A357 Alloy, when the modifier is not added, the eutectic Si usually grows according to TPRE mechanism and precipitate morphology is in the forms of acicular or plate-like, which is for the reason that the preferred orientation of eutectic Si along the <112> Si growth direction [35]. The rare earth Ce is usually preferentially adsorbed in the twin concave groove of the Si phases; the front of the solid–liquid interface after modification effect, due to the coefficient of diffusion of Ce in Al-Si alloy, is small. A proportion of the Ce atoms adsorbed on the surface of eutectic Si is embedded in the Si phases to form the heterogeneous atom defect, resulting in lattice distortion [36], as shown in Figure 6b. This lattice distortion causes Si phases to twin in more directions, prompting Si phases to grow according to the TPRE and change the anisotropy of Si phases growth before modification. The adsorption of Ce atoms at the front of the solid–liquid interface reach a certain degree, that is, the adsorption of Ce on the interface between <110> Si and <100> Al has gained the ability to change the arrangement of atoms on the interface and reduce the lattice mismatch of the two interfaces. The lattice mismatch improves the quality of the Si phase interface lattice,

which promotes the growth phase <110> Si of the Si phases to be epitaxial by being blocked by the Al phase, which makes the preferential growth of the Si phases from <211> to <100> [11,37]. This indicates that Ce atoms have a strong inhibitory effect on Si phases, which make the growth direction change continuously in the process of crystal growth, so that the eutectic Si transformation from coarse and plate-like to fibrous and granular morphology. The fibrous eutectic Si is very imperfect in crystallography, and defects on the surface are likely to cause branching. In this situation, the fibrous Si in the modified alloy allows the alloy to easily bend and split, resulting in a further finer structure. As a result, the morphology and size of the eutectic Si were effectively improved with the addition of Ce. According to the theory of Impurity Induced Twinning, the atomic radius of the modifier is the primary condition for measuring the capacity of modification. The most appropriate ratio is $r_i/r_{Si} = 1.648$ (where $r_i$ is the radius of the modified atom and $r_{Si}$ is the radius of the silicon atom) [38]. The appropriate radius ratio between Ce and Si is $r_{Ce}/r_{Si} = 1.56$ [16]. Therefore, it can be considered that Ce is an effective modifier.

Meanwhile, the Ce atoms can be segregated toward the Si-rich region. SEM results (Figure 10) show that Ce-rich metal precipitates are observed in the vicinity of eutectic Si. These precipitates cause significant undercooling of the melt as they can enhance interfacial energy and poison nucleation sites [39]. As a result, the nucleation of eutectic Si phases is delayed, which leads to the growth of the Si phases being restricted to a certain extent and the refinement of the Si phases.

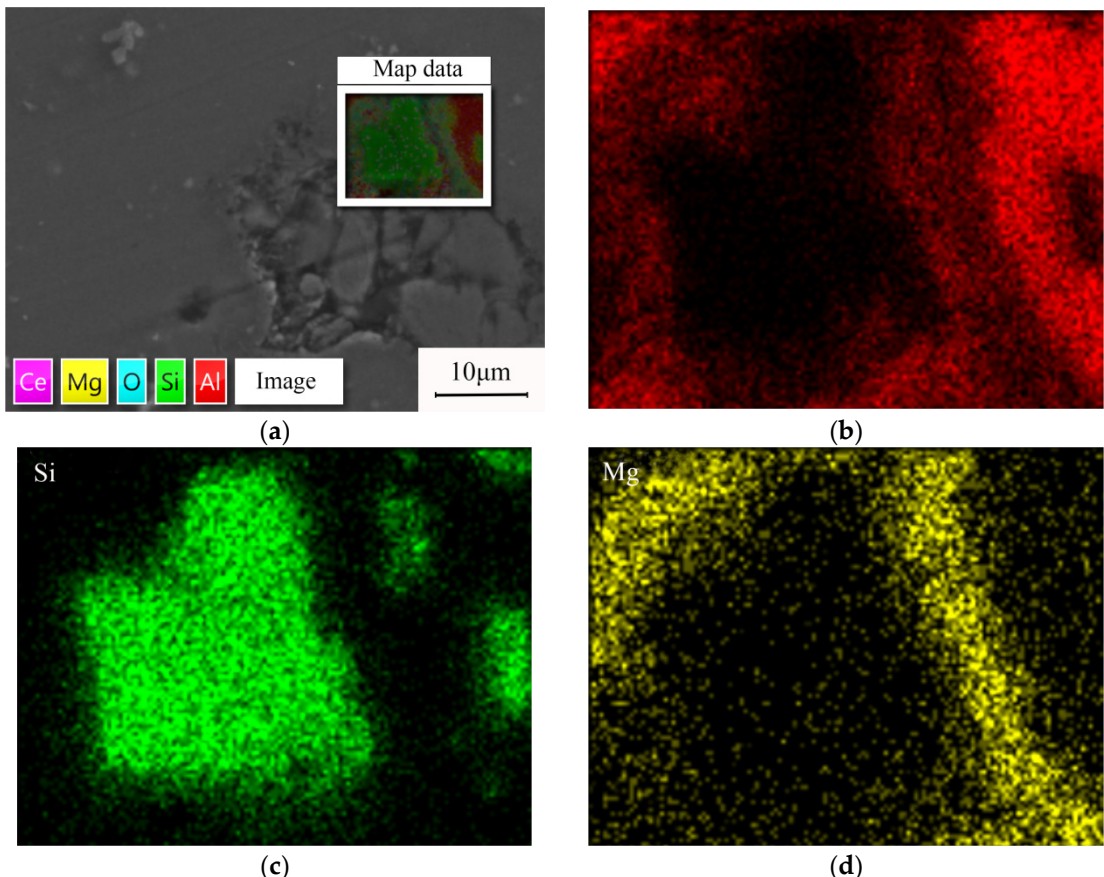

**Figure 10.** *Cont.*

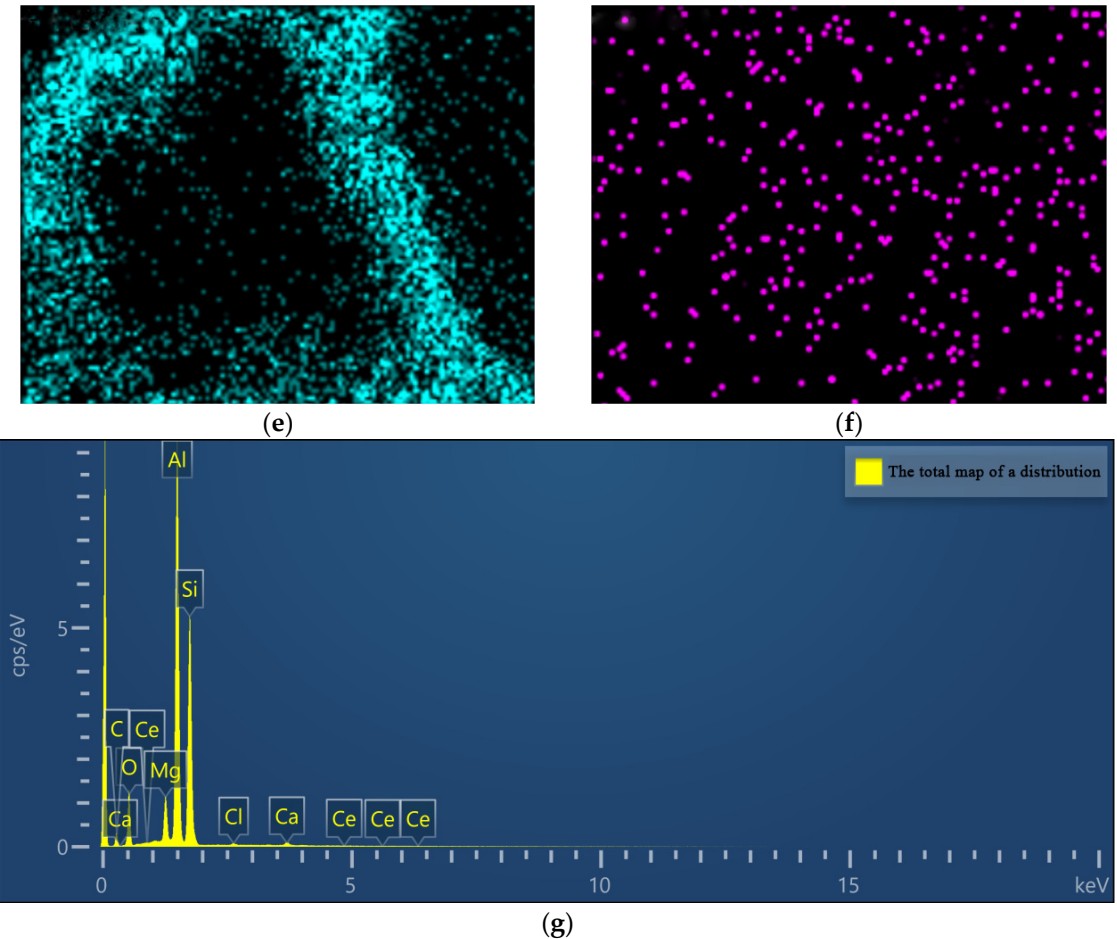

**Figure 10.** Mapping and EDS of Ce-rich metal precipitates. (**a**) Detecting position, (**b**) Al element, (**c**) Si element, (**d**) Mg element, (**e**) O element, (**f**) Ce element and (**g**) EDS analysis.

### 3.3. Effect of Ce on the Mechanical Properties

Figures 11–13 present the values of tensile strength, yield strength, and elongation of industrial A357 alloys with different additions of Ce in as-cast and T6 heat treatment. The results show that the tensile properties of the industrial A357 modified alloy are significantly improved compared with that without addition rare earth alloy. When 0.16% Ce was added, the tensile strength of industrial A357 modified alloy reach 22 0.49 MPa in the as-cast state, and 326.46 MPa in the T6 heat treatment condition, which improved by 22.34% and 18.36%, respectively, compared with the untreated alloy. Nevertheless, the tensile strength is reduced when the content of Ce is greater than 0.16 wt.%.

It can be observed from Figure 12 that with the addition of 0.16 wt.% rare earth Ce to the sample, the yield strength of the industrial A357 alloy under the as-cast and T6 heat treatment conditions is 183.47 MPa and 294.63 MPa, which is 11.79% and 21.55% higher than that of without addition rare earth alloy, respectively. However, the formation of intermetallic compounds will have a harmful effect on the mechanical properties of the alloy when the content of rare earth Ce is high.

From the results of Figure 13, it can be observed that the elongation of industrial A357 modified alloy is significantly higher than those alloys without addition rare earth alloy under both as-cast and T6 heat treatment conditions. With the addition of 0.16 wt.% Ce, the elongation of industrial A357 modified alloy under the as-cast condition is 2.95%, and it reaches 4.59% under the T6 condition, which is improved compared with the as-cast and T6 condition modified alloys 41.15% and 58.82%. In particular, the elongation of the industrial A357 modified alloy did not continue to improve as the content of Ce to increase.

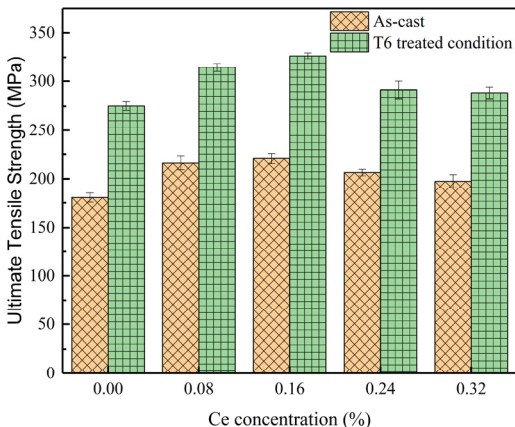

**Figure 11.** Tensile strength of A357 alloy with different Ce content.

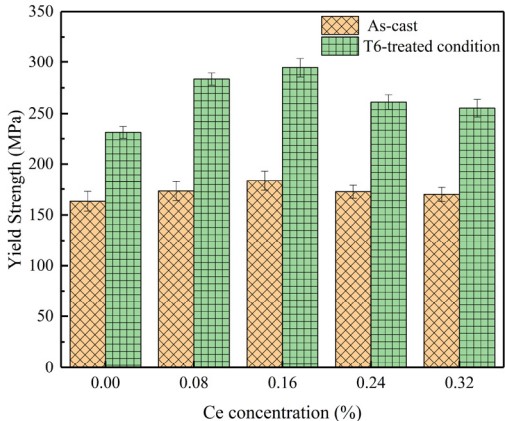

**Figure 12.** Yield strength of A357 alloy with different Ce content.

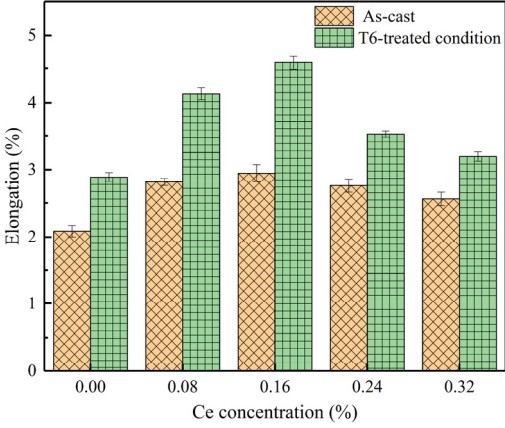

**Figure 13.** Elongation of A357 alloy with different Ce content.

The quality index Q combining strength and toughness was used to evaluate the tensile properties of the alloy because it can better describe the true tensile properties of the casting than the tensile strength or elongation [3]:

$$Q = UST + k \times \log(EL) \qquad (2)$$

Here, *K* is the constant related to the material, *UTS* is the ultimate tensile strength, and *EL* is the elongation. The *k*-value of Al-Si-Mg alloy is ~150 [40]. The resulting value of the quality index is shown in Figure 14, from which we observe that the alloy with 0.16 wt.% Ce added in the as-cast condition have a higher *Q* value of 290.96 MPa, which is about 27.47% higher than that of the without

addition rare earth alloy. Under the T6 heat treatment, the *Q* value when the rare earth Ce was added at 0.16 wt.% was 425.73 MPa, which was about 23.42% higher than that of the without addition rare earth alloy.

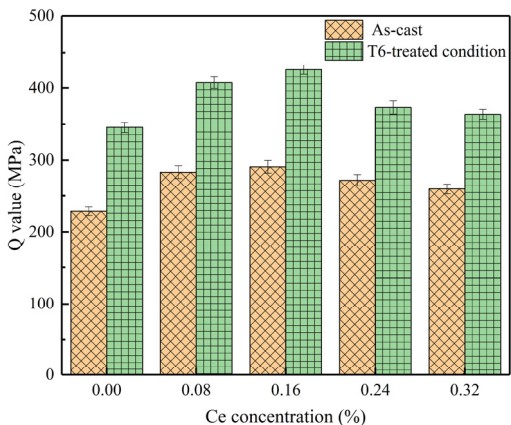

**Figure 14.** *Q* value of the experimental steel treated by different processes.

In general, the mechanical properties of Al-Si casting alloys are mainly affected by the microstructure composition. The grain size was effectively refined when rare earth Ce was added to A357 casting alloy. The improvement of the mechanical properties of the casting is mainly attributed to the fact that the overall grain boundary area will significantly increase with the reduction of the alloy grain size, which can effectively prevent the dislocation movement. At the same time, the morphology of eutectic Si is changed into a fine spherical shape, which weakened the cleavage effect of eutectic Si on Al matrix, and further improved its mechanical properties. In addition, with the addition of rare earth Ce in the modified alloy, the morphology of some β-Fe phases appeared in a spherical shape, as shown in Figure 15. Moreover, when part of the Ce atoms enters the β-phase ($Al_5FeSi$) clusters, some Fe atoms will be replaced by Ce, which causes the needle-like morphology of the Fe-containing intermetallic to be transformed into α-phase (AlSiFeCe)-containing rare earth Ce (Figure 16). The harmful effect of beta on the mechanical properties of the alloy is effectively reduced by the α-phase formed after adding rare earth Ce.

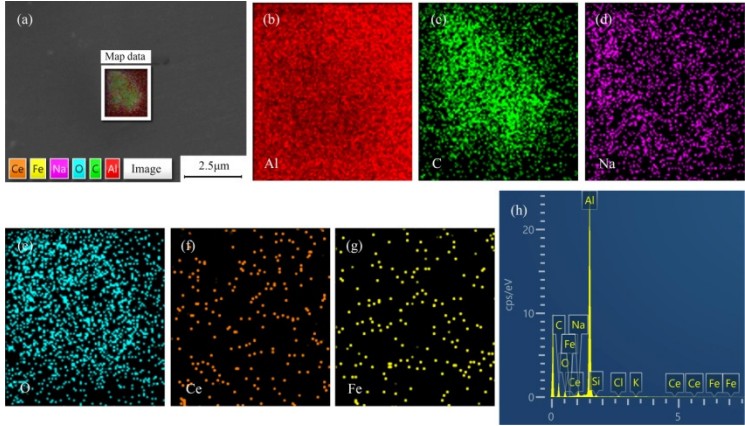

**Figure 15.** (Ce-Fe) phase in A357 alloy. (**a**) Detecting position, (**b**) Al element, (**c**) C element, (**d**) O element, (**e**) Ce element, (**f**) Ce element, (**g**) Fe element and (**h**) EDS analysis.

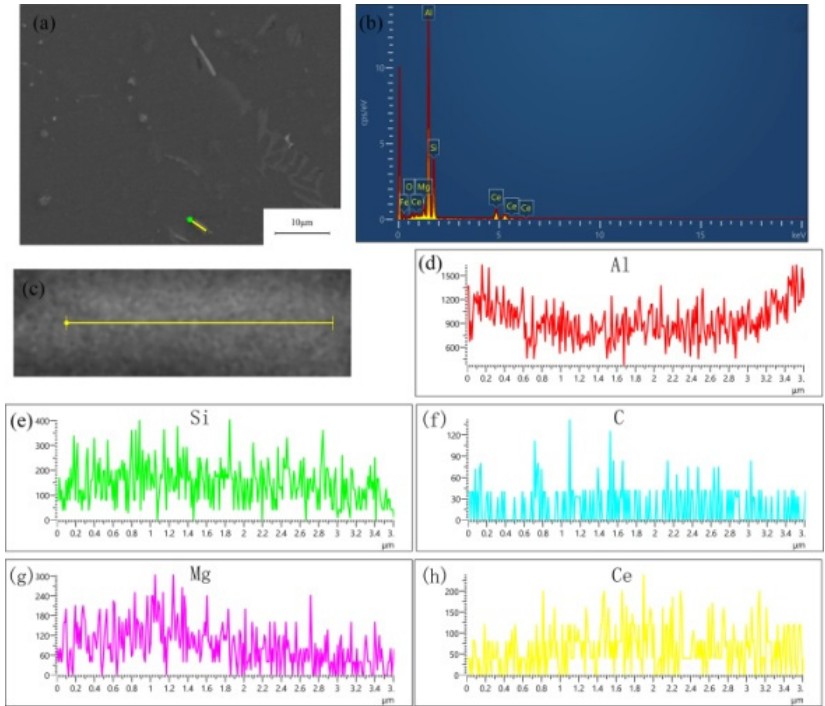

**Figure 16.** (AlSiFeCe) phase in A357 alloy. (**a**) Detecting position, (**b**) EDS analysis, (**c**) Line scanning position, (**d**) Al element, (**e**) Si element, (**f**) C element, (**g**) Mg element, (**h**) Ce element.

However, the mechanical properties of the material decrease when excessive Ce is added. This may be due to the addition of excessive Ce leads to the increase of needle-like morphologies β-Fe size. Meanwhile, some needle-like, flake-like, and plate-like morphology (AlSiCe) intermetallic compounds were observed in the α-Al matrix, as shown in Figure 17.

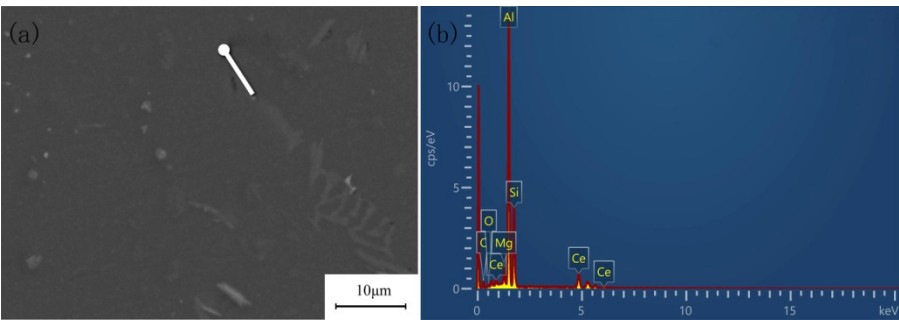

**Figure 17.** Intermetallic compounds of (AlSiCe) in A357 alloy. (**a**) Detecting position, (**b**) EDS analysis.

The nonuniform distribution of Ce in the A357 alloy structure results in the micro-segregation of Ce [41]. The solute redistribution occurs in the crystallization process because of the low solubility of Ce in Al and Si phases. Therefore, most of the Ce exists in the uncrystallized liquid phase, which leads to the low concentration of Ce in the primary phase of the first precipitate. However, the Ce concentration in the residual liquid phase is increased so that it is rich in Ce in the eutectic region of the final crystallization. These (AlSiCe) intermetallic compounds can cause character distortion, grain boundary defects, and seriously affect the microstructure properties of A357 cast alloy [42].

### 3.4. Effect of Ce on the Fracture

Figure 18 shows the SEM fracture morphology of tensile samples of industrial A357 alloy with different Ce additions under T6 condition. It can be observed from Figure 18a that the fracture surfaces

of the tensile specimen of the without addition rare earth alloy under T6 condition display brittle fracture and cleavage planes can be observed, resulting in lower elongation values. The fracture morphology of A357 modified alloy tensile samples containing 0.08 wt.% Ce appear to have a mixed quasi-cleavage and dimple appearance. When the amount of rare earth addition reaches 0.16 wt.% (Figure 18c), the SEM fracture of the industrial A357 modified alloy tensile test specimen presents obvious dimpled fracture morphology, and the dimples have uniform distribution with high density, thus the elongation is significantly increased. As the addition of Ce continues to increase, the SEM display fractographs show deterioration. The EDS spectrum of spectroscopic analysis was performed on the fracture surface of the tensile specimens. The results are shown in Figure 19 and an intermetallic compound (AlSiCe) containing rare earth Ce was observed on the surface of the fracture. It is easy to generate micro-cracks in the place where these compounds exist. When subjected to external forces, micro-cracks will rapidly develop and cause fracture, which seriously affects the mechanical properties of the A357 modified alloy [43].

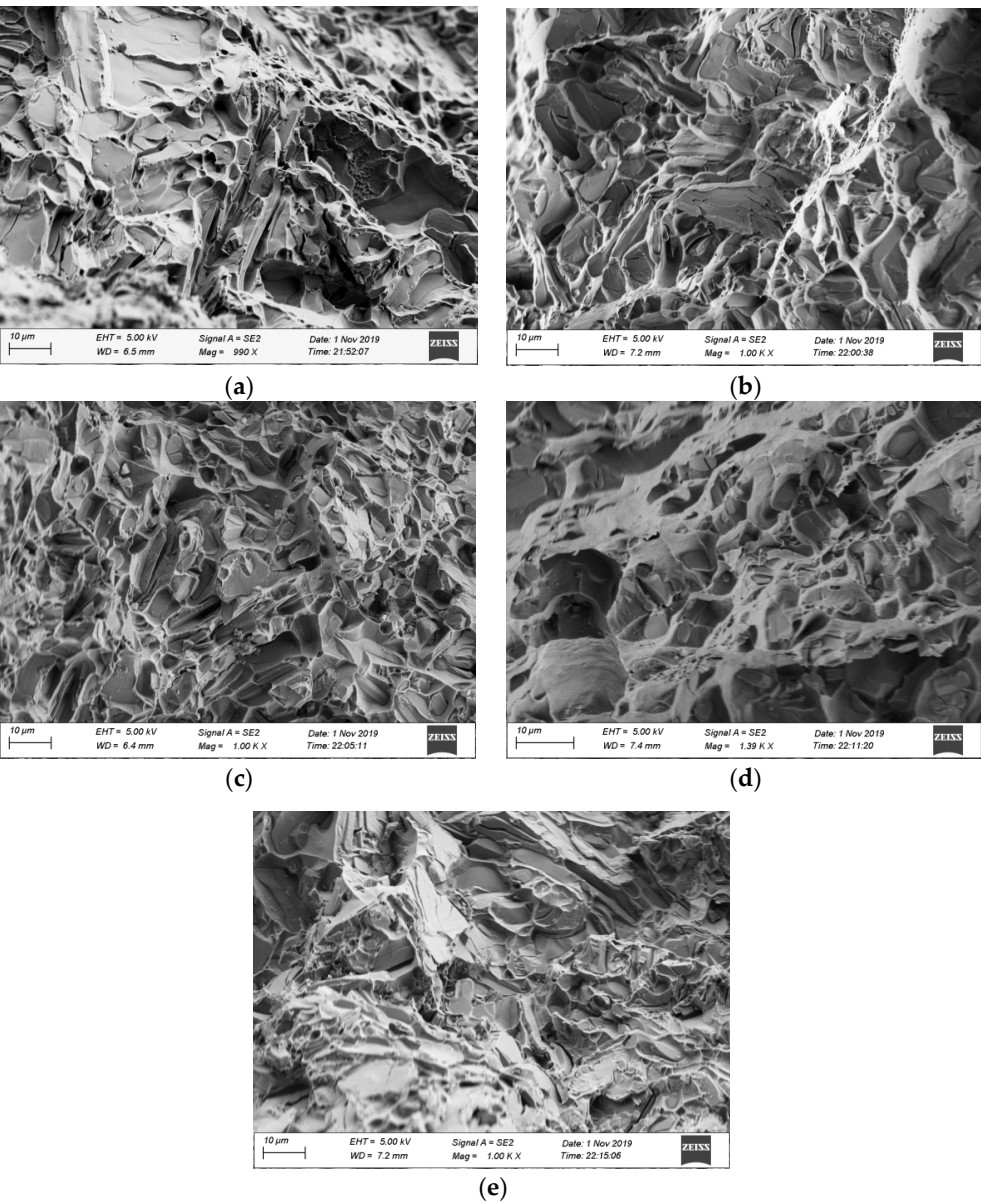

**Figure 18.** Tensile fracture morphology characteristic of the alloys. (**a**) 0% (**b**) 0.08%, (**c**) 0.16% (**d**) 0.24%, and (**e**) 0.32%.

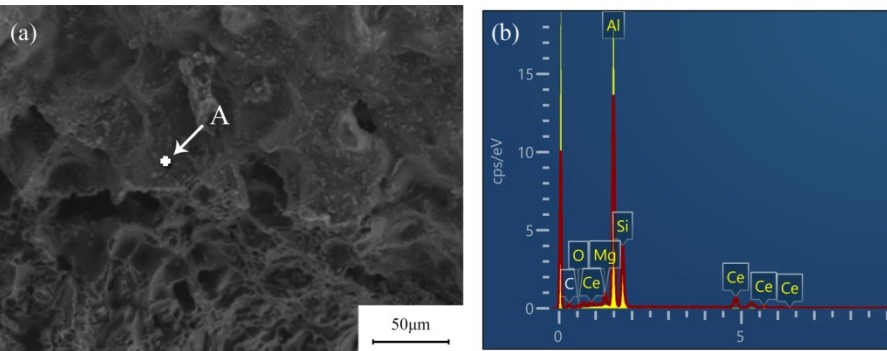

**Figure 19.** SEM fractograph and EDS analysis of the (AlSiCe) intermetallic compound in the A357 modified alloy with 0.16% Ce. (**a**) Detecting position, (**b**) EDS analysis.

It is well known that the fracture process of Al-Si cast alloy consists of three mixing steps: (1) cracking of hard brittle phase, (2) the formation and growth of microcracks, and (3) the expansion of microcracks. To further investigate the fracture mechanism of the modified alloy, Figure 20 is an optical micrograph of the fracture side view of the modified alloy tensile specimens after T6 heat treatment. The main factors affecting the cracking in the first step are eutectic Si particles and SDAS. When subjected to external loading, compared with the α-Al matrix, the stress concentration at eutectic Si is more likely to cause microcracking, and separate from the aluminum matrix, as shown by the arrow in Figure 20a. On the other hand, the fracture path of modified alloy specimens tends to pass through eutectic Si particles during plastic deformation because of the strong interaction between eutectic Si particles and the slip band. In the second step, the cracks tend to grow when eutectic Si particles break down. As the strain increases, some of the adjacent microcracks will coalesce and form large cracks (as shown in the red line in Figure 20a). The third step is related to the local connection of microcrack. The SDAD in modified alloy has smaller value uniformly distributed fine circular eutectic Si particles, which make the grain boundary more discontinuous. Therefore, it is easy to produce a stronger interaction between the slip band and plastic flow in the grain boundary, resulting in higher fracture strain and elongation. Moreover, as shown in Figure 20b, the presence of plate-like (AlSiCe) intermetallic compounds in the modification alloys makes them very prone to generate a longer microcrack. In addition, it is easier to produce cracks at the grain boundary where it intersects with the α-Al primary, which is easily grown and fractured under external loading.

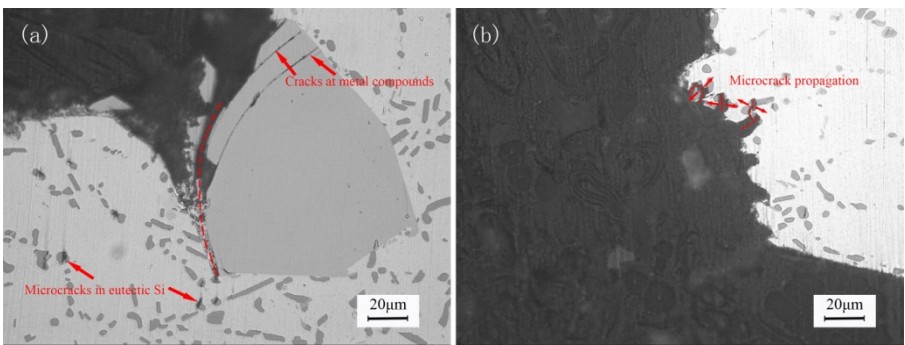

**Figure 20.** Optical micrographs of side views of fractured A357 modified alloy tensile samples. (**a**) Side views 1 (**b**) Side views 2.

## 4. Conclusions

(1) The addition of appropriate rare earth Ce to industrial A357 alloys can cause apparent grain refinement. This is because, after adding rare earth Ce to the alloy, an intermetallic compound containing Ce can be formed as a heterogeneous nucleation point. Moreover, the intermetallic compounds of containing Al, Si, Ce, and Mg elements at the front of the solid–liquid interface

resulting in undercooling of local structure in the alloy liquid. Compared with the unmodified alloy, the addition of Ce reduced the average grain size and SDAS of the alloy in the as-cast and T6 conditions by 50.65%, 42.41%, 54.93%, and 45.99%, respectively.

(2) The addition of appropriate Ce to industrial A357 alloy rarely produces twins. Meanwhile, Ce-rich metal precipitates were observed in the vicinity of eutectic Si. These precipitates can enhance the interface energy and poison nucleation point, which inhibited the growth of eutectic silicon particles to some extent, so that the morphology of the Si phase is improved. In the as-cast and T6 heat treatment conditions, the length of eutectic Si particles in the modified alloy with Ce addition was shortened by 52.49% and 43.12%, and the aspect ratio was reduced by 72.06% and 76.35%, compared to that of the unmodified alloy.

(3) The addition of Ce effectively improves the tensile properties of the alloy compared with the unmodified alloy. The tensile strength, yield strength, and elongation reached 220.49MPa, 183.47MPa, and 2.95%, respectively, in the as-cast condition, and 326.46MPa, 294.63MPa, and 4.59% respectively, in the T6 condition. This is mainly due to the combined effects of grain refinement, eutectic Si modification, and partial β-Fe morphological.

(4) With the addition of Ce, the fracture failure type of the tensile specimen of industrial A357 modified alloy changed from brittle failure to ductile failure, and the fracture surface is mainly dimpled. The presence of the intermetallic compounds containing (AlSiCe) in the modified alloy is prone to cracking and weakens the tensile properties of the industrial A357 modified alloy.

**Author Contributions:** Conceptualization, Y.W. and Z.Y.; methodology, Z.Y.; software, Q.L.; validation, Z.Y.; formal analysis, Z.Y. and Q.L.; investigation, Z.Y. and K.T.; resources, C.Q.; data curation, Z.Y.; writing—original draft preparation, Z.Y.; writing—review and editing, Z.Y.; visualization, Z.Y.; supervision, C.Q.; project administration, Y.W.; funding acquisition, Y.W. All authors have read and agreed to the published version of the manuscript.

**Funding:** This research received no external funding.

**Acknowledgments:** The authors wish to acknowledge C.Q. for support with experimental facilities and preparation. The authors also sincerely thank Xue Su for the technical support provided.

**Conflicts of Interest:** The authors declare no conflict of interest.

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
