# Peer review of "Effect of Ce Addition and Heat Treatment on Microstructure Evolution and Tensile Properties of Industrial A357 Cast Alloy"

_metals, doi:10.3390/met10081100_

Round 1
Reviewer 1 Report
Please see attached file!

Reviewer 2 Report
Submitted manuscript entitled “Effect of Ce addition and heat treatment on microstructure evolution and tensile properties of industrial A357 cast alloy" described in detail effects of addiction Ce on the microstructure, mechanical properties and fracture morphology of industrial A357 cast alloy in as-cast and T6 heat treatment state. Topic is interesting and I recommend it to publication. I think paper is written in good enough language, but maybe English language verification should be done by the right person for English spelling and grammar. Technique, technology and research methods used in the work are adequate. Methods and obtained results prove founded thesis and show originality of the manuscript.
Some small revision of paper is needed.
Table 1 should be change, contents of individual elements are written in two lines, which does not look good, you can stretch this table and then everything will be in one line.
Experimental procedure: page 2, line 90-91, there is written: “The Al-10wt%Ce intermediate alloy with different amounts (0, 0.08, 0.16, 0.24 and 0.32wt %) was added to the molten melt”, it is not clearly written “with different amounts” of what element, of course you can guess that you mean the element Ce, but it not clearly written.
Fig.1 and Fig.2, structure shown on the figures are very small, they look similar, it's hard to see any differences, based on this structures is very difficult to verify if the results shown on figure 3 are true or false.
Figure 4. XRD for which state is it, how much Ce there is? Is there any changes of the diagram for other contents of Ce, and in comparison to state as-cast.
Fig.7 and Fig.8, structures shown on the figures are very small, they look similar, it's hard to see any differences (especially fig.8), base on that figures is difficult to verify what is described in the text – if this is true or false.
Page 11, line 303-307, size of the fonts are twice bigger than in the other text, this text is centered but should be adjust. Also it look that this paragraph is not finished. In line 307 there is a reference to Figure 49, but such a drawing is not in this article, probably it should be fig. 16.
Also in line 307 there is beginning of the next beginning of the next sentence: is written “On th……” and this sentence is cut, please check it, and correct.
Quality of the figure 15 is very poor, it is difficult to read what is marked on (b) diffraction.
On figure 16 there is some white line with dot at the one end, what is it, maybe it should be arrow, which shows places of testing.
Fig.17, very small, blurred, there is even impossible to read what a magnification on figures is.
The literature list (references) is twice bulleted: 1., [1].
Reference 9 is red colour, should be black.
Reviewer 3 Report
The document presents a study of the effect of Ce addition (rare earth) on the microstructure and mechanical properties of A357 Alloy. Although the document is well structured, the English level and novelty are low. Several issues deserve better attention:
- Title: The title should be rewritten. The presented suggestion no make sense, namely what concerns to the heat treatment. An example: “Effect of Ce addition on microstructure and mechanical properties of A357 cast alloy”.
- What is the novelty of the work? The addition of the Ce in the melt aluminum is known since at least 2011.
- According to the authors' suggestions, the main mechanism responsible by refinement of grain and eutectic Si is the “… constitutional undercooling”. However, none graphical information regarding this fact is given. The authors must introduce the cooling curve (T vs t), and the respective derivative. The addition of this information will confirm the suggestion. This information can match well with Figure 5.
- The % of Ce added to the melt seems low. Is there any reason to not test the addition of 1%, for instance. There are other authors that report results for 1% of Ce.
- This alloy presents a high wt% of Ti. As the authors know, the Ti is a refiner. Is there a chance of the Ti be responsible for the refinement of alloy?
- The microstructure present in Fig. 1 and 2 need a better definition.
- Why the grain size of heat treat samples is smaller that is verified for as-cast samples? Strange? Heat treatment doesn´t promote alpha-Al grain refinement.
- Line 149. The first sentence doesn´t make sense. It is a known alloy.
- Table 2: it is suggested to introduce standard deviation.
- Can the Index Q be compared with the works of other authors for A357 Alloy?
- Line 303 to 309: Strange information. Fig.49??
- Figure 14: This figure with this quality and configuration doesn´t present anything. The authors need to be more confident in the information presented. Can be supplied concise graphical information concern the refinement of the intermetallic beta phase?
- All figures (microstructures) must be improved. The quality is poor.
Round 2
Reviewer 1 Report
no comments
Author Response
I found that the Review Report (Round 2) in Reviewer 1 has no comments. The content of the manuscript has not been modified.
Reviewer 3 Report
Although some of the considerations were addressed, there are others that need to be considered. I am not confident about some answers.
- The mechanism responsible by refinement... As the author knows, the operations conditions have a vital impact on the cooling curves. Use references of other authors are not true for the present study conditions. Without cooling curve information, the suggested mechanism is purely speculative.
- The clarifications regarding %Ti as refiner need to be better addressed.
- I am not confident with the answer to the question: “Why the grain size of heat treat samples is smaller that is verified for as-cast samples?”
- “Figure 14 shows the β-phase morphology after Ce modification, which is determined by composition. The experiment is temporarily unavailable due to the COVID-19.”
I can understand, but this issue needs to be discussed and improved.
The document has potential, however it need to be improved/discussed.
Round 3
Reviewer 3 Report
I am continuing to have doubts about grain size diameter before and after heat treatment. No evidence is given to prove that. The suggested document is not clear about it. Also, the issue of the suggested document is different…
